# Infection Spread and High-Resolution Detection of Close Contact Behaviors

**DOI:** 10.3390/ijerph17041445

**Published:** 2020-02-24

**Authors:** Nan Zhang, Boni Su, Pak-To Chan, Te Miao, Peihua Wang, Yuguo Li

**Affiliations:** 1Department of Mechanical Engineering, The University of Hong Kong, Pokfulam Road, Hong Kong 999077, China; zhangnan@hku.hk (N.Z.); ptjchan@connect.hku.hk (P.-T.C.); miaote@connect.hku.hk (T.M.); phwang@connect.hku.hk (P.W.); 2China Electric Power Planning & Engineering Institute, Beijing 100120, China; bnsu@eppei.com

**Keywords:** infection spread and control, infection risk, human behavior, close contact, sensor-based, indoor environment, indoor positioning, head and body motion, open-plan office

## Abstract

Knowledge of human behaviors is important for improving indoor-environment design, building-energy efficiency, and productivity, and for studies of infection spread. However, such data are lacking. In this study, we designed a device for detecting and recording, second by second, the 3D indoor positioning and head and body motions of each graduate student in an office. From more than 400 person hours of data. Students spent 92.2%, 4.1%, 2.9%, and 0.8% of their time in their own office cubicles, other office cubicles, aisles, and areas near public facilities, respectively. They spent 9.7% of time in close contact, and each student averagely had 4.0 close contacts/h. Students spent long time on close contact in the office which may lead to high infection risk. The average interpersonal distance during close contact was 0.81 m. When sitting, students preferred small relative face orientation angle. Pairs of standing students preferred a face-to-face orientation during close contact which means this pattern had a lower infection risk via close contact. Probability of close contact decreased exponentially with the increasing distance between two students’ cubicles. Data on human behaviour during close contact is helpful for infection risk analysis and infection control and prevention.

## 1. Introduction

Indoor human behaviors directly impact on indoor thermal comfort [1], energy efficiency [2], office design [3], and exposure to pollutants (e.g., infectious microbes) [4]. Indoor human behaviors in and between different environments also directly impact on the infection risk [5]. Close contacts are believed to facilitate the spread of many viral respiratory diseases such as influenza [6], SARS [7], MERS [8], and even Ebola [9].

Infection risk via close contact is influenced by interpersonal distance, respiratory activities, and movement of body parts. Interpersonal distance directly affects the risk of virus exposure due to inhalation and deposition, the so-called short-range airborne and large droplet routes, respectively [10]. A threshold distance of close contact less than 1.5 m to 2 m is generally accepted as risky [11,12,13]. Human respiratory activities such as breathing, talking, and coughing can generate droplets of different numbers and sizes [14,15,16,17,18]. Infectious pathogens are shed and exhaled by the infected during these respiratory activities, and transported by the exhaled air streams, while inhalation of fine droplets and exposure to large droplets are also affected by the inspiratory air streams and body/head/arm movement [19]. Relative face orientation (e.g., face-to-face, face-to-side) and posture are important factors in determining the cross-infection, especially over short distance [20]. Exposure of face-to-back close contact is much smaller than it of face-to-face pattern [21,22]. Posture also important in droplet deposition, for example, droplets deposited on trousers on the thighs of a sitting person should be more than on a standing person.

Very little data exist on indoor close contact behaviors, especially data combining all of the factors mentioned above. Some human behaviors are difficult to accurately monitor with high temporal resolution [11,19]. Electronic sensors such as radiofrequency identification devices (RFIDs) are the most commonly used to collect human contact data. These devices, however, can only detect close contact by taking readings of interpersonal distance every 20 s, and only one-on-one close contact can be detected [23,24]. A temporal resolution of 20 s is not sufficient, as the median value duration of a close contact is 17 s [11]. Moreover, human respiratory activities and movement of body parts (e.g., head and body) cannot be detected by this means. Body movements impact the air flows in a room. To collect close contact data with high temporal resolution, video recordings have been used, which can be processed second-by-second [11,19]. This approach is subject to human error on the part of the video analysts, and also time-consuming.

Office are the most common form of workplace for the most of employees, with open-plan designs widely used [25,26]. In this study, we monitored and analyzed indoor human behaviors in a graduate student office using automatic devices installed on the hats and clothes of the participants. These devices overcome nearly all of the shortcomings mentioned above and automatically collect high-resolution data on indoor human behaviors. The data monitored included the indoor position, head and body motion, and posture of each individual which are important for infectious disease transmission. We collected more than 1,440,000 s of data for 49 students across two days. This study supported data of indoor human behaviors on close contact. From indoor positioning distribution, we can know that which part in the office has higher infection risk. Combining individual inhalation and exhalation patterns with head’s and body’s motion during close contact, high-precision quantitative risk assessment on infection spread and control could be conducted.

## 2. Materials and Methods

The experiments were conducted in two close-by Chinese graduate student offices on two consecutive Saturdays. The choice of two nearly identical offices was due to the need of minimising microbial interference, as the experiment also contained another part on surface microbial monitoring, which is not reported in this paper.

### 2.1. Room Setting

Each room was designed for a maximum of 30 students, i.e., containing 30 cubicles (Figure 1). There were 26 students (13 male and 13 female) in Room 1 on the first experiment day (day 1), and 23 students (12 male and 11 female) in Room 2 on the second experiment day (day 2). A total of 32 students participated in the experiment, and 17 of them in both experiments. Among 32 participants, 27 students were those who worked in the institute and five students were invited by some students who were in the institute. In addition, three pairs of students were romantically involved. We selected these students because they were familiar with the experimental environment, which should minimize any bias caused by environment. The length (12.2 m) and height (2.73 m) of both rooms were the same, and the layouts of the cubicles in the two rooms were symmetrical, see Figure 1.

Each room had a water dispenser and a group of lockers. Only Room 2 had a printer. Each room was divided into six cubicle regions (marked by dotted lines in Figure 1), and students in the same cubicle region were more likely to communicate with each other. All of the students were monitored by 22 video cameras (1080P) from 8:30 a.m. to 9:30 p.m. The indoor temperature during the experimental days, which controlled by central air conditionings, was between 26 and 27 degrees. There were 27 fluorescent tubes on the ceiling of the office to keep the luminance, therefore, combing with the high-resolution cameras, most human behaviors can be captured. Each camera monitored one or two office cubicles, except for two with a global view, one camera for the door, and one camera for the water dispenser.

### 2.2. Detection Devices

The sensors we developed are shown in Figure 2a. The sensors for indoor positioning and head motion were installed on the participants’ hats, and those for body motion were installed on the chest of a tight shirt. An ultrawide band (UWB) radio real-time location system (RTLS) was applied to obtain the indoor positions of all participants [27,28]. The distance resolution of UWB is within 0.1 m when there is no obstruction. To ensure continuous data transmission, the UWB tag was installed on the top of each hat, and four UWB anchors were installed on the office ceiling. An inertial measurement unit (IMU), which can measure and record the position and motion of the head and body (i.e., rotation), was also installed on the top of each hat and the front of each tight shirt. A microphone was installed on the collar band of the shirt to determine when the participants were talking. To protect privacy, only the sound level was recorded. Adjustable bands on the hat and around the head were used to avoid relative movement between the hat and head. Tight clothing was worn to avoid relative movement between the shirt and body. All of the data recorded during the experiment were saved in a chip. The weight of a device on hats and shirts was 73 g, and the weight of hat with fixed accessories was 159 g. Therefore, each student worn a hat of 232 g on the head and worn a device of 73 g on the shirt. Light weight brings a much smaller impact on human behaviors.

Rotations were recorded in the form of quaternions. To obtain the absolute (relative to the ground) rotation of the head and body, all of the participants underwent calibration after wearing and before taking off the hat and the shirt. During the calibration, the participant stood still and faced the same wall while keeping his/her head and body upright for 10 s. For any IMU (head or body), the quaternions during the calibrations before and after the experiment were denoted as *q_start_* and *q_end_*, respectively. If the difference between *q_start_* and *q_end_* was greater than a threshold value (equivalent to 10° rotation), it was probable that the participant had moved the hat or shirt during the experiment, and the data were discarded. Data were also regarded as invalid if the fluctuation of rotation between 10-s calibrations was more than 5°. To eliminate drift errors for all sensors, the quaternions were adjusted from all valid raw data based on spherical linear quaternion interpolation (Equation (1)):(1)$$q=q_{raw}q_{start}^{-1}{\left(q_{start}q_{end}^{-1}\right)}^{\left(t-t_{start}\right)/\left(t_{end}-t_{start}\right)},$$
where *t* is time, *t_start_* and *t_end_* are the time of the calibrations before and after the experiment. The steering vectors ($\vec{x_{h}}$, $\vec{y_{h}}$, $\vec{z_{h}}$ for head and $\vec{x_{b}}$, $\vec{y_{b}}$, $\vec{z_{b}}$ for body, relative to the ground) can be obtained from the quaternions (Figure 2b). Usually, the horizontal rotation of the head relative to the body direction is more important. The relative steering vectors of head to body ($\vec{x_{hb}}$, $\vec{y_{hb}}$, and $\vec{z_{hb}}$) can be calculated using coordinate transformation (Equation (2)):(2)$$\left[\vec{x_{hb}}\vec{y_{hb}}\vec{z_{hb}}\right]=\left[\begin{array}{ccc} x_{\vec{x_{b}}} & y_{\vec{x_{b}}\;} & 0\\ x_{\vec{y_{b}}\;} & y_{\vec{y_{b}}\;} & 0\\ 0 & 0 & 1 \end{array}\right]\left[\vec{y_{h}}\vec{y_{h}}\vec{z_{h}}\right],$$
where $x_{\vec{x_{b}}\;}$ and $y_{\vec{x_{b}}\;}$ are the components of $\vec{x_{b}}$ the in x and y directions and $x_{\vec{y_{b}}\;}$ and $y_{\vec{y_{b}}\;}$ are the components of $\vec{y_{b}}$ in the x and y directions, respectively. Here, the roll and pitch of the head are relative to the ground; only the yaw of the head is relative to the body. There is no yaw of the body because no relevant reference point on the waist was monitored. Therefore, data on three head motions (yaw, pitch, and roll) and two body motions (pitch and roll) were collected.

All of the sensors were calibrated prior to the experiments. We first installed the IMUs on a large plate, and performed specific rotations with different angles (e.g., 45°, 90°, 135°, 180°) in three directions. The IMU was regarded as well-calibrated when the difference between all measured values and real values was less than 2°. The UWB sensors were also calibrated in the experimental rooms before the experiments. We chose five indoor points at which to perform calibration to reduce the error of indoor positioning to no more than 10 cm.

### 2.3. Close Contact Behavior

As motions relevant to the inhalation/exhalation flows, we considered individual head and upper body motions, human respiratory activities (e.g., coughing, sneezing, and speaking), and the features of relative position (e.g., interpersonal distance and relative face orientation) of two people. The definitions of head and body motions can be found in Zhang et al. [11].

Close contact was defined as any full or partial face-to-face interaction within 2 m [23,29,30,31]. A face-to-face interaction can occur with or without conversation, including when two individuals read a book or watch a computer screen together. An event was not counted as close contact if the distance between the two students was shorter than 2 m but there was no interaction between them; for example, if two students used their own computers in their own cubicles. If any close contact lasted for more than 1 s, it was counted as a single close contact. If the two students were separated (more than 2 m apart or with no interaction) by more than 1 s, the individual students’ close contact behavior (e.g., with another student at that distance) was counted separately. In this study, interpersonal distance was defined as the distance between the sensors on the two participants other than those on their faces. In addition to interpersonal distance, the relative face orientation angle of the two participants was also obtained. This is the angle between the normal of the two students and ranges from 0° to 180° [11].

### 2.4. Data Processing

During the two experimental days, 1,440,492 s of human behavior data were collected. The first author processed all video episodes second by second, recording all visible close contacts between each pair of students in the office. The collected data include posture (standing, sitting, and squatting), whether two students were in close contact, the identities of those in close contact, and the start and end time of each episode of close contact. The indoor position and head and body motions were monitored by sensors. The indoor position data on *day 1* were unfortunately missed, which means that all of the valid indoor position data were from *day 2*. Out of the 23 participants, 21 had indoor position data during *day 2*, and in total 717,168 s of indoor position data were collected. Among these data, 10,827 s (1.5%) was lost or disrupted, and linear interpolation was used to approximate these data. To maintain the accuracy of all recorded data on head and body motions, we discarded the data for which the difference between two calibrations was more than 10° or the fluctuation during each calibration was more than 5°. After data filtering, 541,200 s of data on head and body motions were valid. Moreover, a total of 1,250,392 s of valid data on sound levels were recorded by the microphones over the two days. All of the results reported below are based on these valid data.

## 3. Results

### 3.1. General Human Behavior Data

While indoors, the students spent 5.3% of their time standing, 94.6% sitting, and only 0.1% squatting. We divided the office into functional areas, i.e., occupied office cubicles, vacant office cubicles, aisle, and areas near public facilities (Figure 3a). Figure 3b shows the distribution of indoor positions during *day 2*, where red indicates the highest coverage rate (≥40 times per 25 cm^2^). Occupied office cubicles had the highest coverage rate, mainly due to occupation by their owner (Figure 3b). The students spent on average 92.2% of their time staying in their own cubicles. For the remaining 7.8% of the time, the students occupied other places. The results show that they spent most of the latter time in the aisle close to the doorway, the area close to the door, and areas close to the printer and the water dispenser (Figure 3c). Some students had a particularly close relationship with certain others or even a romantic relationship (e.g., boy-/girlfriend), and their office cubicles had a higher probability of being occupied by others. Most vacant office cubicles had a very low occupied percentage. In general, when outside their own cubicle, the students spent 47.9%, 4.5%, 37.4%, and 10.2% of their time in occupied office cubicles, vacant office cubicles, the aisle, and areas near public facilities.

Based on 541,200 valid data of head and body motions, we obtained the general characteristics of these motions for the overall group of students. Figure 4a illustrates the head and body motions. The circle shows the probability distribution of face orientation in the form of a projected half sphere; the reader can imagine that the eyes are located at the center of the sphere. This half sphere is divided into 1296 sectors, each of them 5° × 5°. The top, bottom, left, and right correspond to the participant raising, lowering, left-turning, and right-turning the head by 90°. The center indicates that the student is looking almost exactly forward.

This provides an intuitive visualization of the students’ preferred head motions. In the office, the students preferred to look towards the red and orange grids in the circle (i.e., lower their head), and had a very low probability of raising their head (blue grids). To characterize the head motions, we considered the movement of the head in three directions independently. In the horizontal direction, the average degree of yaw was 4.1°, which means that the students on average slightly turned their head to the right by 4.1°.

The students had almost equal probability of turning their head left and right, i.e., the probability distribution of yaw was symmetric. The average degree of pitch was 23.0°, which means that the students preferred to lower their head by 23.0° on average. Indeed, the head was lowered during 93.5% of indoor time. The students on average tilted (rolled) the head 2.8° to the left. The probability distributions of left and right tilts were almost the same. The students spent 86.1% and 97.4% of time tilting their head within 15° and 30°, respectively.

Figure 4b shows the body motions during indoor time. The students on average lowered their bodies (pitch) by 23.2°, and spent 85.6% of indoor time bending their bodies forward. The average degree of body roll was 3.7°, which means that the students on average slightly tilted their bodies to the right. The probability distribution of roll was almost symmetric. The students spent 86.7% and 98.2% of time tilting their bodies within 15° and 30°, respectively.

Three postures were considered in this study: standing, sitting, and squatting. Here we only analyzed head and body motions during standing and sitting because the students spent very little time squatting (0.1%). Figure 5 lists the probability distributions of head and body motions in three directions for all students during indoor time. The rolling of the head and body showed little difference between standing and sitting postures, and the participants had only a slightly higher probability of bending their head during sitting than during standing. However, the characteristics of yaw and pitch of the head, and pitch of the body, differed by posture. The average angles of head yaw during standing and sitting were −3.0° and 4.5°, respectively. The students on average lowered their heads by 30.9° and 24.1° while standing and sitting, respectively. The pitch of the body was strongly linked to posture. The average angles of pitch of the body during standing and sitting were 21.2° and 24.7°, respectively. However, the students had a higher probability of keeping their body bent forward at a large angle during standing (*β_b_* > 60° during 8.1% of the time) than during sitting (*β_b_* > 60° during 0.7% of the time). The most common forward-bending angle of the body during standing was between 10° and 15°, while that during sitting was 35° to 40°.

### 3.2. Indoor Behavior During Close Contact

The students spent more than 9.7% of their time in close contact and each student had on average 4.0 close contact episodes per hour. The probability distribution of close contact fitted a log-normal distribution (Figure 6). The average and median durations of close contact were 54.5 and 15 s, respectively. Close contacts with duration between 8 and 16 s had the highest frequency. The durations of 38.2%, 68.8%, and 82.8% of close contacts were no more than 10, 30, and 60 s, respectively. From the microphone data, at least one student was speaking during 68.6% of the time during close contact. One-on-one close contacts accounted for more than 90% of close contact time, while 7.8%, 1.5%, and 0.6% of close contact time involved three, four, and five students simultaneously. Six-student conversations only accounted for 0.06% of close contact time.

During close contact, 22.9%, 76.3%, and 0.8% of students stood, sat, and squatted, respectively (Table 1). During 59.6% of the close contact time, both students were sitting. The pattern of one standing and one sitting was adopted during 33.5% of the close contact time. The students almost never chatted with each other while one stood and one squatted (0.1% of close contact time).

Figure 7 shows the characteristics of close contact. As can be seen, 68.2% of the close contacts were between students in the same region, and most were between adjacent students. Only 13.2% of the close contacts were between students in remote regions (Figure 7a). Influenced by the circulation of people, students near the aisle and the door had a higher probability of being contacted (Figure 7b). From Figure 7c, the greater the distance between the office cubicles of two students was, the lower was the probability of a close contact. The average cubicle distance between two students for close contact was 1.25 areas (for the area distribution refer to Figure 3a), and 77.2% of close contacts occurred between two students with a cubicle distance of no more than three areas. The student who had the closest contacts had 50.5 episodes per hour (Figure 7d). On average each student had 4.0 close contact episodes per hour, and no one had zero close contacts during either day. During the two days, 12.2% (total pairs of contact: 141; possible pairs of contact: 1156 = 26 × 25 + 23 × 22) of all possible pairs of students had close contact. Each pair of students had close contact on average 11.3 times per day, and the pair with the highest frequency of close contacts had 100 episodes per day (Figure 7e). As shown in Figure 7f, 22.4%, 40.8%, 59.2%, and 85.7% of the students spent no more than 1%, 5%, 10%, and 20% of their time in close contacts, respectively. The most sociable student spent 34% of her indoor time in close contact. The average and median ratios of close contact time to total indoor time were 9.7% and 6.4%, respectively. During the two days, the students had close contacts with an average of 8.4 and 4.2 students, respectively. The most sociable student had close contacts with 10 and 20 students during *day 1* and *day 2*, respectively (Figure 7g).

From Figure 7h, the times of day with the most students in the office were between 9:30 and 11:30 and between 13:00 and 17:30. The peak frequency of close contact was between 11:00 and 12:30 and after 16:00.

Figure 8 shows the interpersonal distance by posture and gender. The average interpersonal distance during close contact was 0.81 m. The average interpersonal distances between sitting-sitting, standing-standing, and standing-sitting students were 0.74 m, 0.93 m, and 0.88 m, respectively. There were three peaks of interpersonal distance during close contact between two students who were sitting. As shown in Figure 8a, the first peak (0.1–0.3 m) was caused by pairs of participants with a very close relationship (e.g., boy-/girlfriend), the second peak (0.6–0.7 m) was caused by pairs of students who sat back-to-back, and the third peak (1.1–1.2 m) corresponded to the distance between two adjacent students. The probability distributions of interpersonal distance for sitting-standing students and standing-standing students accorded with log-normal distributions, and the most frequent interpersonal distances were 0.5 and 0.7 m, respectively. As 60% of close contact was between two sitting students, the overall probability distribution of interpersonal distance (black line) was similar to that of two sitting students.

From Figure 8b, 21.4%, 22.8%, 38.1%, and 17.7% of close contacts were between two male students (M-M), two female students (F-F), a male and a female student (non-couple) (M-F_NC), and couples (M-F_C). The contact rates (ratios of actual relationships with close contact to total possible relationships) for M-M, M-F, and F-F were 15.7%, 25.9%, and 10.3%, respectively. The average episodes of close contact per day between each pair of M-M, M-F_NC, M-F_C, and F-F who had contact with each other were 6.9, 8.2, 70.5, and 12.1, respectively. The average durations per close contact between each pair of M-M, M-F_NC, M-F_C, and F-F were 68.3, 48.7, 53.6, and 58.5 s, respectively. The average interpersonal distances during close contact between these four groups were 0.88 m, 0.70 m, 0.96 m, and 0.71 m, respectively. There was no interpersonal distance shorter than 0.2 m during close contact between two male students. Two female students preferred close contact at a short distance (0.1–0.4 m). Students of different genders (non-couple) had two peaks of interpersonal distance, which were related to their seating position. Couples had a normal distribution of interpersonal distance, and preferred distances between 0.4 and 0.6 m.

There were three major posture patterns during close contacts in the office: sitting-sitting (both students sitting), standing-standing (both students standing), and sitting-standing (one student sitting, the other standing). The head and body motions of students and relative face orientation angles between students under different patterns of posture during close contact are shown in Figure 9. When two students were both standing or sitting, they preferred to look slightly downward. The average pitch angles of the head under sitting-sitting and standing-standing were 14.7° and 20.1°, respectively (Figure 9a,b). The average pitch angles of the body under these two conditions were 24.3° and 17.8°, respectively. However, under the sitting-standing condition (Figure 9c), the eye direction of the standing student was much lower than that of the sitting student.

The average pitch angles of the head of the sitting and the standing students in a sitting-standing pattern were 11.3° and 34.0°, respectively, and the average pitch angles of the body were 22.2° and 33.9°, respectively. The relatively low probability of looking downward for the sitting student implies that the standing student was usually located at the side of the sitting student rather than face-to-face. Standing students had a very high probability of facing downward.

Two sitting students had a high probability of only a slight relative angle of face orientation (5° to 25°) during close contact, which means that they usually faced in similar directions. Two standing students preferred face-to-face close contact with a relative face orientation angle between 150° and 170°. There was no obvious preference regarding relative face orientation angle under the sitting-standing pattern.

## 4. Discussion

### 4.1. Automatically Collected Indoor Human Behavior

In this study, we provided the first comprehensive dataset combining the indoor position, head and body motions, and posture of students at the same time which are helpful for infection risk assessment via close contact route. Indoor human behavior is strongly dependent on the type of indoor environment. In a hospital ward, patients usually lie in their beds, while health care workers walk between rooms and beds [32]. In an aircraft cabin or a cruise ship, passengers usually sit in their seats, while crew members walk through the aisle to provide service and food [33,34,35]. Both passengers in an aircraft cabin or a cruise ship and students in an office spend most of their time in their own seats or in aisles. Children in nurseries have been found to spend more time standing than sitting during indoor free-play time [36]. In a primary school, pupils have been found to spend 2.1 times more time sitting than standing during school hours [37]. However, in this study of a graduate student office, the students spent 94.6% of their time sitting. Therefore, people may tend to prefer sitting with increasing age, although the type of environment is also highly influential.

Close contact is an important activity in daily life, and also plays a critical role in infectious disease transmission. The duration of close contacts directly determines the exposure to viruses. Many researchers have reported the distribution of the duration of close contact (Figure 10) in different types of indoor environment. However, RFIDs or wireless sensors can only detect close contacts at intervals of 20 s [24,31,38,39,40,41], an insufficient temporal resolution for close contacts, which have a median duration of only 17 s. Although video observations can reach a temporal resolution of 1 s [11], the subjectivity of video analysts and the huge workload are two major shortcomings. As summarised in Figure 10, all types of indoor environment show similar values of the cumulative probability distribution (CDF) of the duration of close contacts. Brief close contacts (<20 s) are dominant, and prolonged close contacts (>300 s) are rare. Conferences and museums have a higher rate of long close contacts, while in hospitals and congress buildings shorter close contacts are more common. In a video observation study of a graduate student office, the average and median duration of close contact were 53.8 s and 17 s [11], respectively. Our study, meanwhile, found the average and median duration to be 54.5 s and 15 s, respectively.

### 4.2. Close Contact Behavior

We collected and analyzed three types of data during close contact: indoor position, head and body motion/movement, and posture. These three factors are important to infectious disease transmission. Indoor position can help calculating the interpersonal distance of people during close contact. The infection risk decreases sharply with the increase of the interpersonal distance [12,20,42,43]. Posture, and head and body movement influence the body plume during close contact. For example, short-range exposure can be affected strongly by body plumes [44]. Frequent movement of the head and body during conversation can change not only the orientations of the exhaled/inhaled airflows, but also the patterns of body convective flows and the thermal plume. The exhaled airflows of two people also interact with and affect each other [12]. Various gestures involving small movements of the hands, palms, legs, eyebrows and other small-scale facial features may not significantly affect the body plume or exhaled flows [45,46]. Posture also important in droplet deposition. For example, more droplets may deposit on thighs of a sitting person if he/she talks with an infected. People have high probability to touch their thighs and legs with the frequency of more than 30 times per hour [19]. It may lead to a high infection risk because people also have high touch frequency on mucous membranes [47]. Relative face orientation is a critical factor for exposure during close contact, and it could be calculated by body and head motion. Previous studies found that the exposure during face-to-face close contact is the most, followed by face-to-side pattern, while face-to-back pattern had the lowest exposure [21,22].

As no such data on indoor human behavior in an office had previously been published, we mainly compared the sensor-collected data from this study with those of our previous experiment based on video observation [11] (Table 2). As head and body motions cannot easily be recorded using observation methods, the accuracy is difficult to guarantee. This may be the cause of the disagreements between the results in Table 2. In addition, in our previous experiment, all participants were students whose cubicles were in the studied office. In this study, however, some students were from the next room and some were invited into the office under study by students who worked there. Therefore, the various relationship networks may have caused the difference in the characteristics of close contact and of indoor human behaviors during close contact.

Our previous experiment showed that students spent 9.9% of their time in close contact interactions, while in this study that percentage was 9.7% ± 8.8%. The probability distributions of the students’ posture during close contact differed between the two experiments, although sitting was always the most common posture, followed by standing. Almost none of the students had close contact when squatting. Sitting-sitting and sitting-standing were the most common posture patterns during close contact. In this study, the sitting-sitting pattern was much more dominant than in the previous experiment because all students chose their office cubicles before the experiment. Students with the closest relationships sat next to each other, and could chat without standing up or walking. During both experiments, one-on-one close contact accounted for almost 90% of all close contacts. The number of participants per close contacts depends on the type of indoor environment. For example, more people will participate in a close contact in a group discussion, while one-on-one close contact is very highly probable in a doctor’s consulting room.

Preferred interpersonal distances differ by culture, ethnicity, relationship, personal habits, age, gender, and ambient environment [48]. Closer interpersonal distance means higher virus inhalation which increases infection risk via close contact route [42]. The average personal distances for acquaintances and intimately close persons in China are 83.6 and 57.6 cm, respectively [49]. However, there is no clear definition of interpersonal distance. Our previous study defined the interpersonal distance as that between two mouths, while this study defined it as the distance between two sensors worn on the top of the head. The previous study showed the average interpersonal distance in the office to be 0.67 m. In this study, the average interpersonal distance (0.81 m) was larger because of the different definitions of this parameter. Both experiments found the same effect of gender on interpersonal distance: two female students adopted the shortest distance while students of different genders (non-couples) were farthest apart. As an office is a public area, intimate couples do not interact at such close distance as they would otherwise. Posture also influences the interpersonal distance. Both experiments showed that two sitting students had the shortest interpersonal distance. In this study, more than 50% of close contacts were between adjacent or back-to-back students, and the probability distribution of interpersonal distance was strongly influenced by their cubicle positions. A compact indoor design would lead to shorter interpersonal distance during close contact.

The circulation of people around the office strongly influenced the probability of close contacts. Students at long distances apart had low probability of contact with each other. Therefore, a linear office cubicle design would lead to a lower close contact frequency compared with a matrix design because of longer average distance. People are generally disinclined to make close contact with someone a long distance away if there is no critical information to communicate. In this study, most close contacts were between students sitting in the same region because students with good relationships had chosen to sit together. Most students in this office were acquaintances, and had high probability of a brief close contact when encountering each other. Therefore, the students near the aisles, especially those near the door, had a high close-contact frequency with remote students (Figure 7b) as a result of office circulation. There was no close contact between 88% of the possible pairs of students (connection rate = 12%). In general, this value is strongly related to the level of intimacy between people indoors. For example, the connection rate tends to be very high in a home, but low in public environments such as conference rooms and aircraft cabins. This may explain why homes have a higher infection risk than other environments during an infectious outbreak [29].

This sensor-based study recorded higher pitch values of the head and body, which showed that students preferred to lower their head and lean their body forward. When sitting, they were usually looking at a computer monitor (mostly laptop computers) or reading a book from a desk, and therefore both the torso and head leaned forward. Over the long term, such a sitting posture may lead to a high prevalence of kyphosis among students [50]. Research has also showed that tilting the head forward by 15° places about 27 pounds of force on the neck [51]. This increases to 40 pounds at 30°, 49 pounds at 45°, and 60 pounds at 60° [51]. The damage caused by untreated ‘text neck’ can be of similar severity to occupational overuse syndrome or repetitive stress/strain injury. The office management team could be encouraged to educate their workers and provide ergonomic office equipment such as laptop stands to promote proper posture.

In simulations of infectious disease transmission via the close contact route, face to face is the most common orientation assumed, and all heads are assumed to be at the same height [12,20]. However, in our analysis, the students spent relatively little of their close contact time in a face-to-face position. Staring at the same screen, reading the same paper, or chatting without making eye contact were also common. Before this study, no data had been published on the percentage of time spent speaking during close contact. We found that students in close contact on average spent 68.6% of their time speaking. Studies have shown that talking for 5 min can generate the same number of droplet nuclei as a cough, i.e., some 3000 droplet nuclei [10], and speaking usually involves prolonged exhalation [52]. Our data may provide a reference for the simulation of infectious disease transmission via speaking during close contact. Moreover, combined with indoor surface touching behaviors [19], a comprehensive simulation of infection spread in an office considering the long-range airborne, fomite, and close contact routes could be performed [29]. The results would provide support for infection risk analysis on a large scale such as a city [5,53].

### 4.3. Limitations

This study has several limitations. The presence of the cameras might have had a psychological impact on the students’ behaviors. The experimental hats and shirts showed slight relative movement when the participants moved their heads and bodies. Although we discarded some data with very large fluctuations, some error remained (resolution: 5°). Still, this error was much smaller than that of observation methods. Another limitation is that the interpersonal distance was defined as the distance between two sensors installed on the top of the experimental hats rather than between two mouths or noses. The interpersonal distances obtained in this study were longer than those between two mouths/noses, which are normally used to simulate close contact between two persons. Our experiment collected more than 1440,000 s of indoor data and more than 139,000 s of close contact data over 2 days. This data volume is large, but still insufficient to represent all indoor human behaviors in graduate student offices. Moreover, the close contact behaviors presents the characteristics in the office with around 25 students. Building type and total indoor population will influence the human behaviors.

## 5. Conclusions

Students spent long time on close contact (9.7%) in the office, which may explain the importance of the close contact route for many respiratory infections. The probability of close contact decreased exponentially with the increasing distance between two students’ cubicles. Therefore, students who sit closer to the infected student, have much higher infection risk via close contact route than students who sit further. Comparing with pairs of sitting students, pairs of standing students had lower infection risk via close contact route because they did not prefer a face-to-face talk. The fact that standing students much preferred lower their head and body than sitting students may lead to a shorter distance of exhalation jet than we thought.

## Figures and Tables

**Figure 1 ijerph-17-01445-f001:**
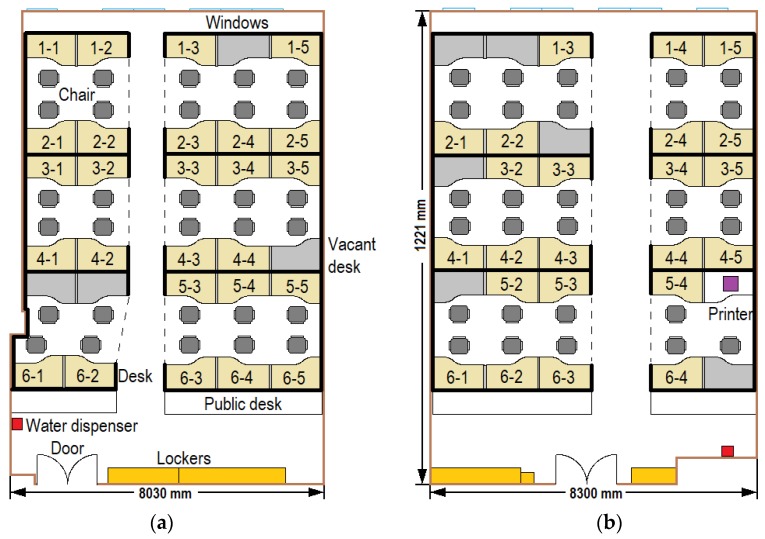
Room settings: (**a**) Room 1; (**b**) Room 2.

**Figure 2 ijerph-17-01445-f002:**
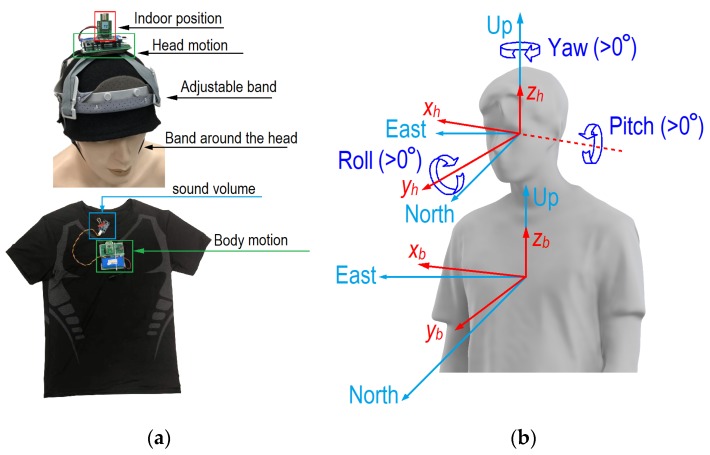
Device design and motions of head and body. (**a**) Device for indoor positioning, and head and body motion detection; (**b**) current and base vectors for head and body.

**Figure 3 ijerph-17-01445-f003:**
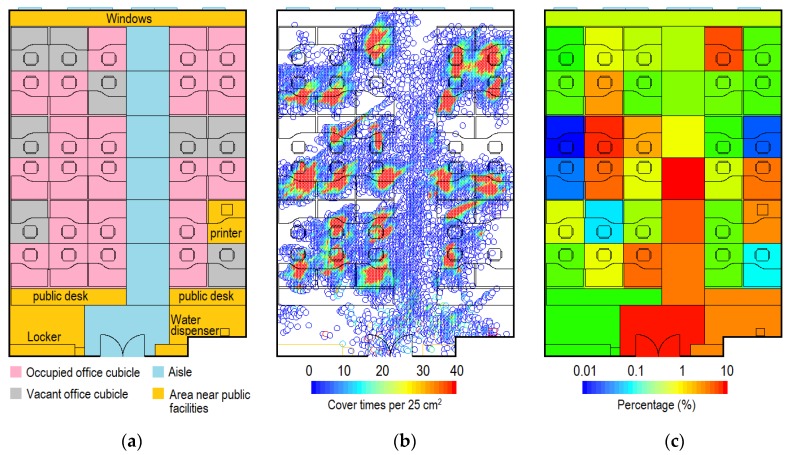
Indoor positioning in Room 2 (*day 2*). (**a**) Functional area; (**b**) Distribution of indoor positions; (**c**) Distribution of indoor positions by functional area during the time spent in other places (i.e., students outside their own cubicles). (Indoor positioning data for students 3-4 and 3-5 were lost).

**Figure 4 ijerph-17-01445-f004:**
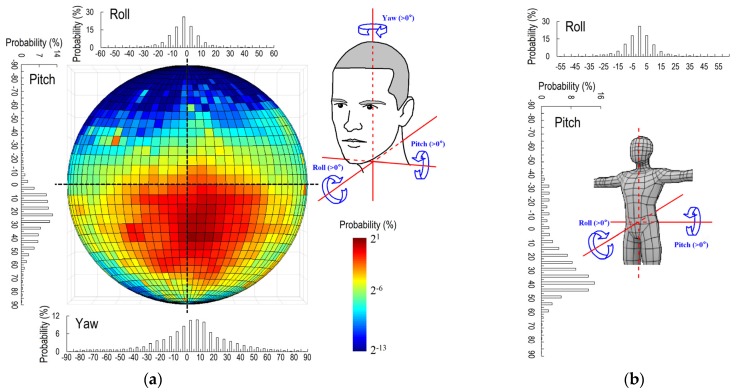
Indoor human behavior in terms of head and body motion. (**a**) Head motion; (**b**) Body motion. (Yaw of head is relative to the body; pitch and roll of head and all body motions are relative to the ground).

**Figure 5 ijerph-17-01445-f005:**
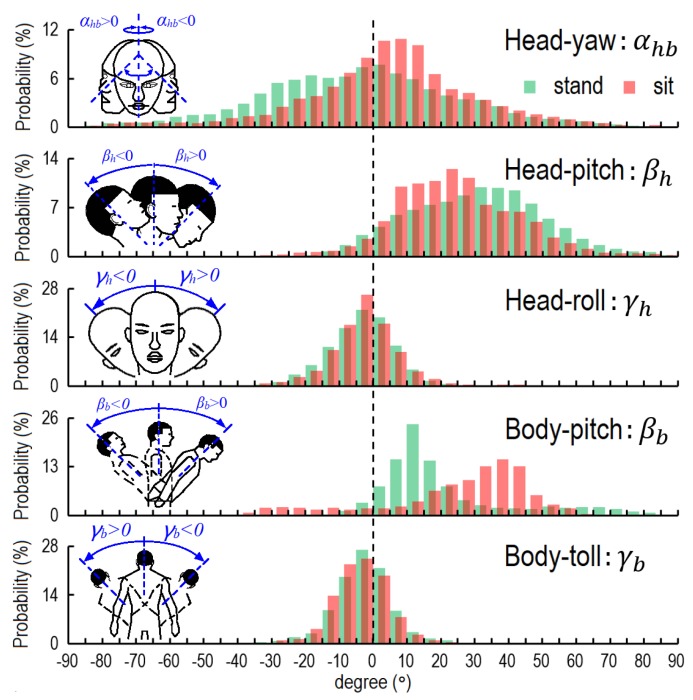
Probability distribution of head and body motions by posture.

**Figure 6 ijerph-17-01445-f006:**
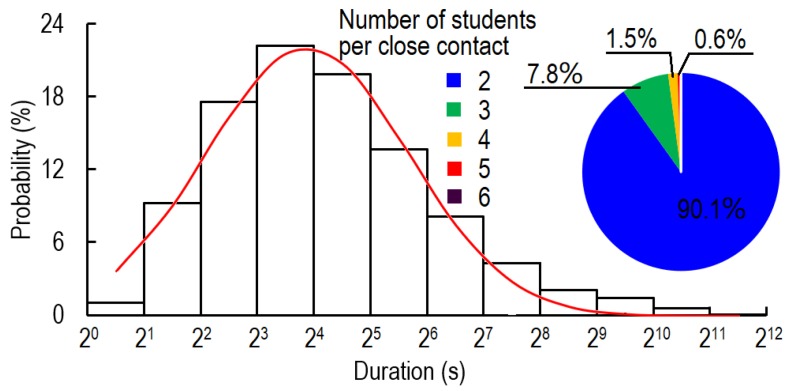
Probability distribution of duration per close contact.

**Figure 7 ijerph-17-01445-f007:**
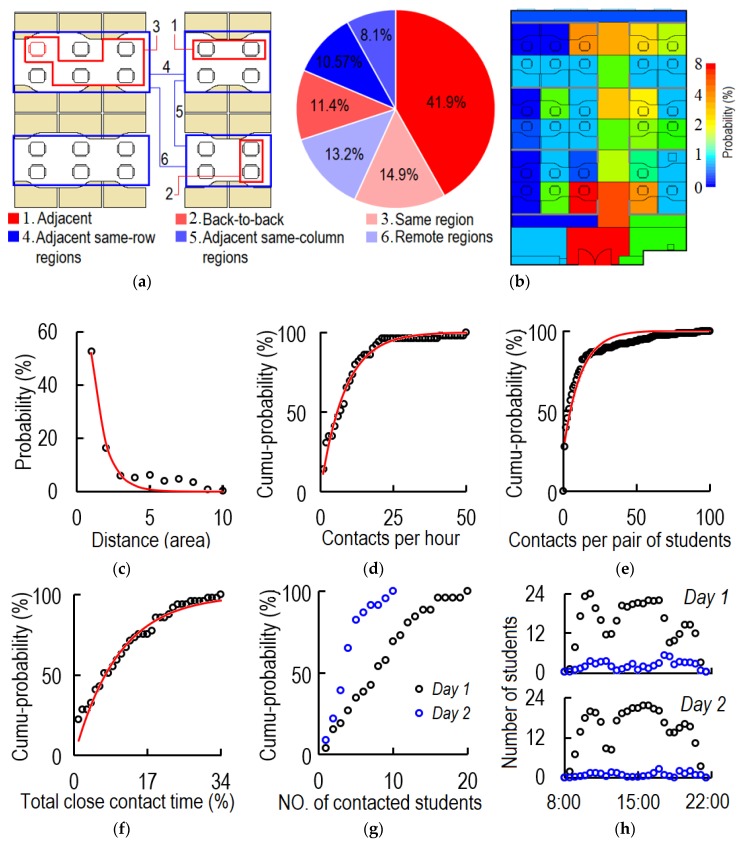
Characteristics of close contact. (**a**) Probability distribution by relative position of students’ office cubicles (same region means two students are in the same region but not adjacent or back to back; percentage shows the episodes of close contact occurred in different relative positions); (**b**) Probability of area occupancy by area during close contact between remote students (students in different regions). The colour bar shows the ln values; (**c**) Probability distribution by distance between work cubicles of two students (distance is calculated as number of functional areas between cubicles of the two students, as illustrated in Figure 3a. For example, distance = 1 for adjacent or back-to-back office cubicles, and distance = 5 between *cubicles 2-2* and *3-2* (see Figure 1b and Figure 3a)); (**d**) Cumulative probability distribution by frequency of close contact (episodes/hour); (**e**) Cumulative probability distribution by total episodes of close contact per day between each pair of participants (episodes/day); (**f**) Cumulative probability distribution by ratio of close contact time to total indoor time (%); (**g**) Cumulative probability distribution by number of contacted students per day; and (**h**) Distribution of number of students who stayed in the room and had close contact during *day 1* and *day 2* (black and blue points are total number of indoor students and total number of students in close contact, respectively).

**Figure 8 ijerph-17-01445-f008:**
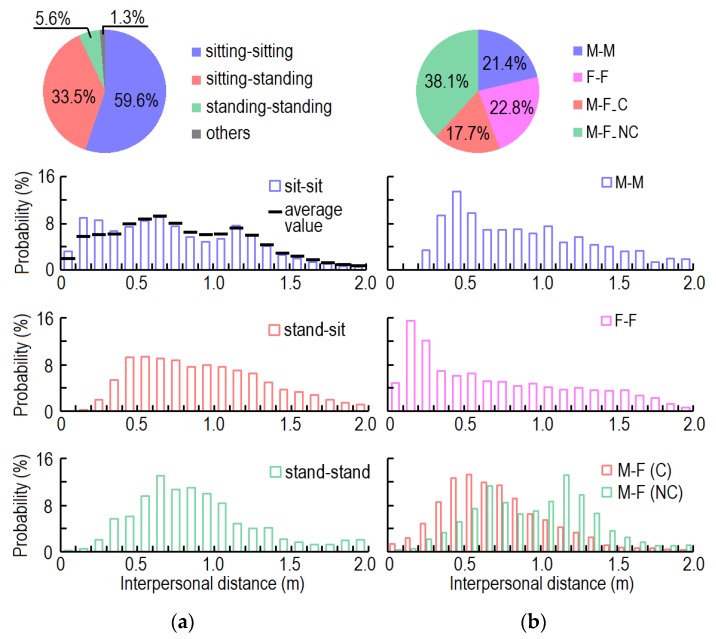
Probability distribution of interpersonal distance by: (**a**) Posture; (**b**) Gender.

**Figure 9 ijerph-17-01445-f009:**
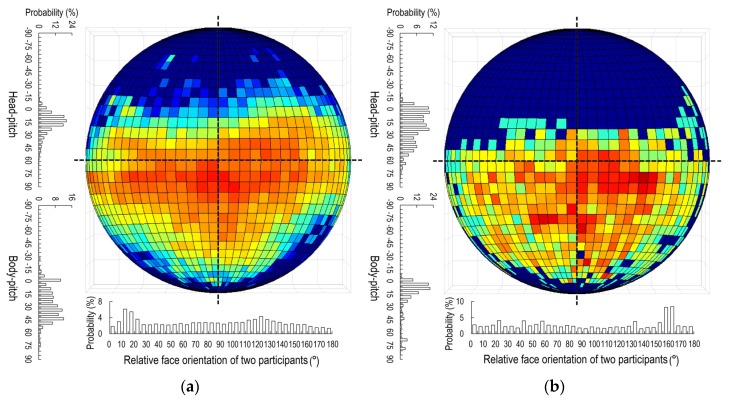
Head and body motions during close contact by posture of the two students: (**a**) Sitting-sitting; (**b**) Standing-standing; (**c**) Sitting-standing. (The circle shows the face orientation of the student; see Figure 4a).

**Figure 10 ijerph-17-01445-f010:**
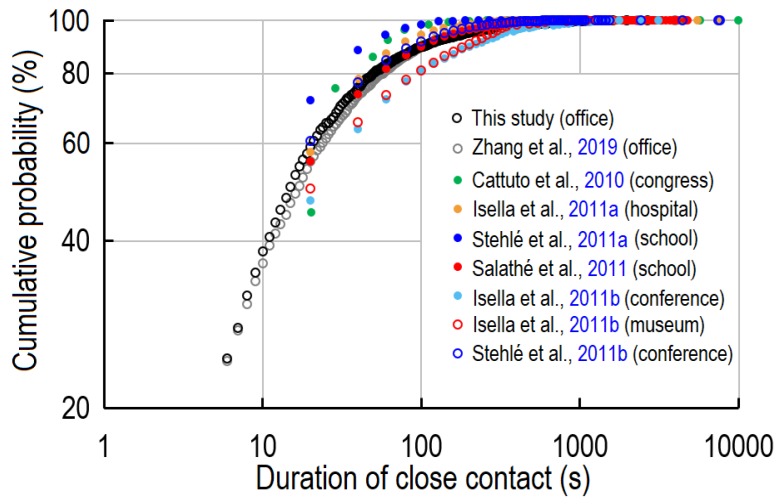
Cumulative probability distribution (CDF) of duration per close contact.

**Table 1 ijerph-17-01445-t001:** Probability distribution of individual postures and posture pattern of students during close contact.

Description	Posture	Percentage
Individual posture	Sitting	76.3%
	Standing	22.9%
	Squatting	0.8%
Posture pattern	Sitting-sitting	59.6%
	Sitting-standing	33.5%
	Standing-standing	5.6%
	Sitting-squatting	1.2%
	Others	0.1%

**Table 2 ijerph-17-01445-t002:** Comparison of indoor human behaviors during close contact in a graduate student office between the previous observation study and the present sensor-based study.

Parameter.	Description	OB ^1^	SB ^2^	Parameter	Description	OB ^1^	SB ^2^
Individual posture	Sitting	59.5%	76.3%	Average ID ^3^		0.67 m	0.81 m
	Standing	40.4%	22.9%	ID by gender	M-M	0.66 m	0.88 m
	Squatting	0.1%	0.8%		F-F	0.55 m	0.70 m
Posture pattern of two students	Sit-sit	34.3%	59.6%		M-F (NC)	0.79 m	0.96 m
	Sit-stand	46.2%	33.5%		M-F (C)	0.58 m	0.71 m
	Stand-stand	18.3%	5.6%	ID by posture	Sit-sit	0.59 m	0.74 m
	Others	1.2%	1.3%		Sit-stand	0.72 m	0.88 m
Number of students per close contact	2	87.5%	90.1%		Stand-stand	0.70 m	0.93 m
	3	11.1%	7.8%	Head/body motion	Head (yaw)	0.9°	2.3°
	4	1.2%	1.5%		Head (pitch)	10.3°	17.9°
	5	0.1%	0.6%		Body (pitch)	10.6°	25.3°
	≥6	<0.1%	<0.1%		Body (bend)	0.1°	3.7°

^1^ Observation study: all data obtained through observation of video tapes by video analysts.11 ^2^ Sensor-based study (this study): all data obtained by sensor detection. ^3^ ID: interpersonal distance.

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
