# Peer review of "Infection Spread and High-Resolution Detection of Close Contact Behaviors"

_ijerph, 2020, doi:10.3390/ijerph17041445_

Round 1
Reviewer 1 Report
This manuscript presents an important field assessment study of human behavior, which are crucial for determining the risk of cross infection in modern indoor environment. The authors developed a novel wireless system to track the position and posture of occupants. The system has good accuracy, but it seems to need better stability to minimize data loss in future measurements.
The authors adopted this system in a student office with rather high occupancy. They found the most frequent locations, average contacts per hour and average mutual distances, etc. during contacts. These data provide solid evidences for future epidemiology studies.
I have several questions and suggestions regarding to the message delivering in the methodology part.
It could be helpful to briefly introduce the indoor environment of the office, including the operative temperature, humidity, lighting, in section 2.1. It is important to eliminate the bias of the environment and its influence to human behavior. How heavy is the device? Could the weight affect the head posture in the study? When two students were close for 1 sec with certain interaction, it was counted as one contact. How to determine this critical time, and what parameters does it depend on? In page 4 line 144, you wrote ‘the first author processed all video episodes second by second, recording all visible close contacts between each pair of students in the office.’ The author should explain the nature of ‘processed’ by, e.g. describing the processing time for each second. Otherwise it is a bit beyond reality to process 1,440,492 pieces of data by hand.
Author Response
Reviewer #1
General comments:
This manuscript presents an important field assessment study of human behavior, which are crucial for determining the risk of cross infection in modern indoor environment. The authors developed a novel wireless system to track the position and posture of occupants. The system has good accuracy, but it seems to need better stability to minimize data loss in future measurements. The authors adopted this system in a student office with rather high occupancy. They found the most frequent locations, average contacts per hour and average mutual distances, etc. during contacts. These data provide solid evidences for future epidemiology studies. I have several questions and suggestions regarding to the message delivering in the methodology part.
Response:
Thanks for your suggestions. Our responses are listed below:
Comments 1:
It could be helpful to briefly introduce the indoor environment of the office, including the operative temperature, humidity, lighting, in section 2.1.
Response:
The indoor temperature was controlled by central air conditionings, and it was between 26 and 27 degrees during the experimental days. However, we did not measure the relative humidity in the office. There are 27 fluorescent tubes on the ceiling of the office to keep the luminance. We have also added this description into section 2.1. The added part is listed below:
“The indoor temperature during the experimental days, which controlled by central air conditionings, was between 26 and 27 degrees. There were 27 fluorescent tubes on the ceiling of the office to keep the luminance, therefore, combing with the high-resolution cameras, most human behaviors can be captured.”
Comment 2:
It is important to eliminate the bias of the environment and its influence to human behavior. How heavy is the device?
Response:
The device was 73 g, and the hat with the fixed accessories was 159 g. Therefore, each student worn a hat of 232 g on the head and worn a device of 73 g on the shirt. The weight is not very heavy, and it will bring a very small impact on human behaviors. We have also added this part into the manuscript.
Comment 3:
Could the weight affect the head posture in the study?
Response:
The weight including hat and device is 223 g. The weight of a human head is around 2500 g. Therefore, the weight is only one-tenth of the heads’. In addition, we also did several pre-experiments ourselves. We think there is no obvious affect on head posture during the experimental day.
Comment 4:
When two students were close for 1 sec with certain interaction, it was counted as one contact. How to determine this critical time, and what parameters does it depend on?
Response:
The time resolution of the device is 1 sec. Therefore, we only can get the interpersonal distance data every second.
Comment 5:
In page 4 line 144, you wrote ‘the first author processed all video episodes second by second, recording all visible close contacts between each pair of students in the office.’ The author should explain the nature of ‘processed’ by, e.g. describing the processing time for each second. Otherwise it is a bit beyond reality to process 1,440,492 pieces of data by hand.
Response: The first author processed the video using the multiple time speed. If a close contact was detected, the speed was changed to the normal speed to log the real information of per close contact including start point, end point, interpersonal distance, and posture. It really took a lot of time.

Reviewer 2 Report
General comments:
This is a very interesting paper on human behaviors during close contact. The author used sensors on hats and shirts to capture human indoor position, posture, and head/body motion during close contact. The obtained data is useful for infection assessment and control via close contact route. The data resolution is very high and data volume is large. I recommend that the paper should be accepted after following questions to be solved.
Minor comments:
Relationship between students play a critical role in close contact pattern. What the relationship between students in the office? Did all students are invited into the office for the experiments? Or they belonged to the office before the experiment? The author should explain that how they hired these students. The author mentioned that students spent 92.2%, 4.1%, 2.9%, and 0.8% of their time in their own office cubicles, other office cubicles, aisles, and areas near public facilities, respectively. However, the most important part of the paper is human behavior of close contact. In Figure 3c, the author listed the distribution of close contact. I recommend that the author had better listing the detailed spatial distribution of close contact other than the spatial distribution during all indoor time. In Figure 3a, there are eight vacant office cubicles, but in Figure 1b, there are only six cubicles. Why two more cubicles are shown in Figure 3a? In Figure 5, the author had better listing a legend to show the direction of the head and body movement. It will be helpful for reader’s understanding. In Figure 5, why did sitting students have higher pitch of body than standing students? In Figure 7a, what does the numbers 1 to 6 mean? In Figure 7a, does the percentage show the close contact episodes or total duration of close contact? The author should give a clarification. In Figure 7b, the color bar should be upside down. The author studied close contact in two rooms. I think the total number of students will influence the results of the close contact. For example, if an office only has two people other than 20 people, what the difference will be? In Figure 9c, why many standing students looked downward with a bit of left head turn during close contact?
Author Response
Reviewer #2
General comments:
This is a very interesting paper on human behaviors during close contact. The author used sensors on hats and shirts to capture human indoor position, posture, and head/body motion during close contact. The obtained data is useful for infection assessment and control via close contact route. The data resolution is very high and data volume is large. I recommend that the paper should be accepted after following questions to be solved.
Response:
Thanks for your suggestions. Our responses are listed below:
Comment 1:
Relationship between students play a critical role in close contact pattern. What is the relationship between students in the office?
Response:
Most of the students were colleagues in the same institute, which means they knew each other before the experiment. Some students were invited to the experiment by their friends who worked in the office. Some students were not only colleagues but also lovers. Among 32 participated students, 27 students were those who worked in the institute and 5 students were invited by some students who were in the institute. In addition, there were 3 pairs of lovers. We have also added this description into the manuscript.
Comments 2:
Did all students are invited into the office for the experiments? Or they belonged to the office before the experiment? The author should explain that how they hired these students.
Response:
All students in the institute were invited, but only 27 students in the institute participated the experiment. There totally were three offices in the institute, and all these 27 students came from these three offices. Two offices were selected for our experiment. We hired these students because they were so familiar with the experimental environment. And it will minimize the bias brought by the environment. We have also added the explanation to the manuscript.
Comments 3:
The author mentioned that students spent 92.2%, 4.1%, 2.9%, and 0.8% of their time in their own office cubicles, other office cubicles, aisles, and areas near public facilities, respectively. However, the most important part of the paper is human behavior of close contact. In Figure 3c, the author listed the distribution of close contact. I recommend that the author had better listing the detailed spatial distribution of close contact other than the spatial distribution during all indoor time.
Response:
The indoor positioning distribution during close contact strongly depends on the relationship between students. For example, if students A and B have a good relationship, The areas around them will have many episodes of close contact. Therefore, there is little significance for indoor positioning analysis during close contact.
Comment 4:
In Figure 3a, there are eight vacant office cubicles, but in Figure 1b, there are only six cubicles. Why two more cubicles are shown in Figure 3a?
Response:
In Figure 1b, six cubicles are vacant. Figure 3a shows the indoor positioning information. Unfortunately, indoor positioning data for two participants (Students 3-4 and 3-5) were lost. And cubicles of these two students are regarded as vacant only in indoor positioning analysis. We have added an explanation into the captain of Figure 3a.
Comment 5:
In Figure 5, the author had better listing a legend to show the direction of the head and body movement. It will be helpful for reader’s understanding.
Response:
We have added the legend to show the direction of the head and body movement. The revised figure is listed below:
Comment 6:
In Figure 5, why did sitting students have higher pitch of body than standing students?
Response:
Standing students usually lowered their heads other than lowered their body during close contact.
Comment 7:
In Figure 7a, what does the numbers 1 to 6 mean?
Response:
We have added the explanation of numbers 1 to 6 to Figure 7a. The updated Figure 7a is listed below:
Comment 8:
In Figure 7a, does the percentage show the close contact episodes or total duration of close contact? The author should give a clarification.
Response:
Percentage shows the episodes of close contact occurred in different relative positions. We have added the explanation into the caption of Figure 7a.
Comment 9:
In Figure 7b, the color bar should be upside down.
Response:
We have revised the color bar in Figure 7b.
Comment 10:
The author studied close contact in two rooms. I think the total number of students will influence the results of the close contact. For example, if an office only has two people other than 20 people, what the difference will be?
Response:
This is really a limitation. Some results depend on total population in the office such as frequency of close contact. In order to clarify this problem, we have added a limitation in to the end of Discussion. The added sentences are listed below:
“Moreover, the close contact behaviors presents the characteristics in the office with around 25 students. Building type and total indoor population will influence the human behaviors. ”
Comment 11:
In Figure 9c, why many standing students looked downward with a bit of left head turn during close contact?
Response:
This related to the relative position between two students in close contact. When sitting students communicated with standing students, standing students had higher probability to stand at the right of the sitting students. Therefore, in Figure 9c, sitting students preferred turning their head right and upper, while standing students preferred turning their head left and downward.

Reviewer 3 Report
This manuscript presents a data collection mechanism for studying the close contacts within two open office areas. Both video recording and wearable sensors were used to study the close-contact activities related to both frequency and postures. Although the study presented in this manuscript were well-studied in several prior works of literature with very similar objectives and general approaches, the experiments conducted in this study does have some interesting new ideas. Besides, the near real-time data collections could provide a more in-depth understanding of the person-to-person close-contacts. However, the reviewer is not clear how the results of the study are relevant to the infectious disease transmissions despite the general knowledge on close contacts are related to the disease transmissions. Besides, the critiques that the authors presented in the introduction section regarding existing literature are not addressed in this study. Therefore, the reviewer cannot recommend the publication of this manuscript on the International Journal of Environmental Research and Public Health in its present form. Below, several issues and questions the reviewer has for the authors to address before the next submission.
Item 1. In section 1, Authors mentioned that human respiratory activities, movement of body parts, and body movements related air flows in the room are the primary deficiencies in the current literatures, however, other than the movements of the body parts, these issues were not addressed in the study of the current manuscript. Even though the reviewer agrees that these issues are essential for infectious disease spreads, and with better temporal resolution, these issues can be more carefully explored. But the reviewer cannot find a thorough study on these issues, and how that relevant to the increase of temporal resolution?
Item 2. At the end of Section 1, the reviewer recommends the authors add a summary of the research contributions and how these contributions closely related to disease transmissions?
Item 3. Section 3.2 It is not clear for the reviewer why the postures (e.g., stand, sit, squat) of the individuals are relevant to the disease transmissions? Besides, the reviewer fails to understand how body and head orientations related to disease transmissions? For example, two person's face-to-face close-contact, sneezing, coughing may have more direct correlations to the transmission of viral respiratory diseases. However, the reviewer cannot make the connections between the elements under study related to these well-known close contact activities.
Item 4. The reviewer disagrees the results presented in Figure 8 are relevant to the disease transmissions. Please clearly explain why the authors believe that, for example, the sitting-sitting position would have better or worse chances for contracting a disease than the standing-standing position? Or the close contacts between genders could have correlations with the disease transmissions? Can an individual's body posture patterns (e.g., body angels or facial orientations) have correlations to the probabilities of contracting a disease?
Item 5. In Section 4.2, the authors attempt to connect the results from this study to general infectious disease transmissions. The arguments are vague and too general, and therefore, not convincing. The reviewer suggests that it might be better to focus the connections to a particular disease or a type of disease. Also, the connections between the results of this study to some disease transmission parameters that generally understood.
Item 6. Table 2. Page 13, The results observed via videotapes and senor detections are significantly different. Some of the observations have 200% differences. How can the authors draw some consistent conclusions for this study? Can these discrepancies suggest inconclusive results? Or are these discrepancies due to the measuring errors? Authors may want to have clear explanations on which observations are trustworthy and which ones are not, and why the authors can draw reliable and consistent conclusions under such volatile results?
Author Response
Reviewer #3
General comment:
This manuscript presents a data collection mechanism for studying the close contacts within two open office areas. Both video recording and wearable sensors were used to study the close-contact activities related to both frequency and postures. Although the study presented in this manuscript were well-studied in several prior works of literature with very similar objectives and general approaches, the experiments conducted in this study does have some interesting new ideas. Besides, the near real-time data collections could provide a more in-depth understanding of the person-to-person close-contacts. However, the reviewer is not clear how the results of the study are relevant to the infectious disease transmissions despite the general knowledge on close contacts are related to the disease transmissions. Besides, the critiques that the authors presented in the introduction section regarding existing literature are not addressed in this study. Therefore, the reviewer cannot recommend the publication of this manuscript on the International Journal of Environmental Research and Public Health in its present form. Below, several issues and questions the reviewer has for the authors to address before the next submission.
Response:
In general, we believe that the reviewer may not be familiarize with the mechanisms of the transmission of disease during close contact, i.e. exposure to the expired air stream (exhaled jet or puffs, in technical term intermitted jet). Body and head movement, and their relative location determines the exposure to the expired jet of the infected. This study provide data of indoor human behaviors on close contact, which is helpful for infection spread research. From indoor positioning distribution, we can know that which part in the office has higher infection risk. Combining with individual inhalation and exhalation patterns, head and body movement during close contact also provide useful data for infection risk analysis. Many current studies hypothesized face-to-face close contact for simulation because there is no data to show the postures and relative positions between two people during close contacts. We support a detailed data on posture, relative position, and head/body movement during close contact, and fill the gap for human behavior on close contact. The results are also helpful for understanding the mechanism of close contact.
Comment 1:
In section 1, Authors mentioned that human respiratory activities, movement of body parts, and body movements related air flows in the room are the primary deficiencies in the current literatures, however, other than the movements of the body parts, these issues were not addressed in the study of the current manuscript. Even though the reviewer agrees that these issues are essential for infectious disease spreads, and with better temporal resolution, these issues can be more carefully explored. But the reviewer cannot find a thorough study on these issues, and how that relevant to the increase of temporal resolution?
Response:
This study got the head and body’s motion during close contact. Head’s motion directly determines the exhaled jet. Relative position, which determined by interpersonal distance and relative head’s and body’s motion of two people, strongly influence the infection risk. In addition, although we did not study how the head’s and body’s motion influence the air flows in the room, we supported the data for those who can work on it. Previous studies used RFIDs to detect close contact have the temporal resolution of approximately 20 s, which are very coarse and may be not suitable for supporting high-precision quantitative risk assessment. We have improved it to 1 s in our study. The obtained results are helpful for mechanism and infection risk analysis of close contact route.
Comment 2:
At the end of Section 1, the reviewer recommends the authors add a summary of the research contributions and how these contributions closely related to disease transmissions?
Response:
We have added a summary at the end of Introduction to demonstrate the contribution to infectious disease transmission. The added part is listed below:
“This study supported data of indoor human behaviors on close contact. From indoor positioning distribution, we can know that which part in the office has higher infection risk. Combining individual inhalation and exhalation patterns with head’s and body’s motion during close contact, high-precision quantitative risk assessment on infection spread and control could be conducted.”
Comment 3:
Section 3.2 It is not clear for the reviewer why the postures (e.g., stand, sit, squat) of the individuals are relevant to the disease transmissions? Besides, the reviewer fails to understand how body and head orientations related to disease transmissions? For example, two person's face-to-face close-contact, sneezing, coughing may have more direct correlations to the transmission of viral respiratory diseases. However, the reviewer cannot make the connections between the elements under study related to these well-known close contact activities.
Response:
Posture is an important factor in infection spread via close contact. First, posture decides the breathing air flow. When a standing person watch the same screen with a sitting infected person, the thermal plume of the sitting infected person will influence the particle inhalation of the standing person. Second, body and head orientation decides the relative horizontal and vertical angles between two people who are in close contact. For example, if a sitting person having a close contact with a standing person, the sitting person preferred rising his/her head and standing person preferred lowering the head. The large droplets from standing person are easily to deposit on the mucous membranes. Moreover, body orientation also played an important role in infection spread. For example, if an infected person talked with a susceptible person with a face-to-face and body-to-body position, large droplets released by the infected person will deposit on the front of the body of the susceptible person. And if an infected person talked with a susceptible person with a relative angle of 90 degrees, most droplets will deposit on the side of the body of the susceptible person. Therefore, head’s and body’s motions are critical for disease transmission.
Comment 4:
The reviewer disagrees the results presented in Figure 8 are relevant to the disease transmissions. Please clearly explain why the authors believe that, for example, the sitting-sitting position would have better or worse chances for contracting a disease than the standing-standing position? Or the close contacts between genders could have correlations with the disease transmissions? Can an individual's body posture patterns (e.g., body angels or facial orientations) have correlations to the probabilities of contracting a disease?
Response:
Figure 8 shows the probability distribution of interpersonal distance by posture and gender. Interpersonal distance is one of the most important factor in infection spread. Distribution of interpersonal distance is changed by posture and gender. Sitting-sitting students had the different distribution for interpersonal distance with standing-standing students, and it will influence the infection risk a lot. In addition, as we mentioned above, droplet deposition between two sitting students is different with that between two standing students. Gender also influence the interpersonal distance distribution. We found that two females had the shortest average interpersonal distance, while a female and a male had the longest one. This fact definitely influences the infection risk.
Comment 5:
In Section 4.2, the authors attempt to connect the results from this study to general infectious disease transmissions. The arguments are vague and too general, and therefore, not convincing. The reviewer suggests that it might be better to focus the connections to a particular disease or a type of disease. Also, the connections between the results of this study to some disease transmission parameters that generally understood.
Response:
In this study, we mainly support the data for human behaviors on close contacts, which are useful for infection spread research. Our objective is to support these data for other researchers for their study on some specific types of disease. Many diseases transmit via close contact route such as influenza and SARS. We do not think it is necessary to make a specific connection to a particular disease or a type of disease, because the type of disease does not change persons’ contact behaviors significantly. Moreover, the main objective of this paper is providing the data on close contact behaviors, not on the transmission of a certain disease.
Comment 6:
Table 2. Page 13, The results observed via videotapes and senor detections are significantly different. Some of the observations have 200% differences. How can the authors draw some consistent conclusions for this study? Can these discrepancies suggest inconclusive results? Or are these discrepancies due to the measuring errors? Authors may want to have clear explanations on which observations are trustworthy and which ones are not, and why the authors can draw reliable and consistent conclusions under such volatile results?
Response:
Two studies were based on different methods, and it will bring some differences. In observation study, 3D postures are reconstructed based on 2D videos by video analysts. The results are subjective and may be inaccurate due to the limitation of videos’ resolution and analysts’ spatial perception ability. Sensor based study’s results are obtained by sensors and therefore are objective, with no human errors. Therefore, we think that sensor based study are more reliable comparing to traditional observation study. Moreover, the difference on close contact behaviors is large. For example, students may stand more if there are more discussion during the day. Therefore, posture distribution had more differences between two studies. Distributions of number of students per close contact obtained from two studies are similar. The obtained results for interpersonal distance by gender from two studies are also consistent. Both studies showed that a male student and a female student (non-couple) had the longest interpersonal distance during close contact, then two males, couples, while two females had the shortest interpersonal distance. The absolute values from two studies on interpersonal distance had some errors because the definitions of interpersonal distance in two studies are not same. We also mentioned the limitation in the study. The limitation showed that “Our experiment collected more than 1440,000 s of indoor data and more than 139,000 s of close contact data over 2 days. This data volume is large, but still insufficient to represent all indoor human behaviors in graduate student offices.”.

Round 2
Reviewer 3 Report
The reviewer fails to see significant improvements in the current form of the manuscript, other than some narrowly opinionated arguments without the supports of reliable references or evidence. The reviewer does believe his comments may not be 100% accurate or unbiased, but the authors can easily establish his/her arguments supported by existing literature works or published historical data.
Starting your responses by referring the reviewer with "In general, we believe that the reviewer may not be familiarized with the mechanisms of the transmission of disease during close contact,...," may not be an intelligent move to get positive reviews on your manuscript. The reviewer happens to have more than 20 years of research and practical experience in state and local government agencies. Attack on the reviewer's credentials may not be your best approach to respond to the reviewer's comments.
In general, the reviewer's comments are aimed to have the authors making connections, in the manuscript, between the research results to relevant disease transmissions. The reviewer failed to see enough statements or explanations on these aspects, and respectfully disagree with some of the relevance the previous manuscript claimed. The reviewer was expecting more professional arguments or proof of relevancy with support from the published works of literature and data in this revision, instead of groundless claims.
The authors' responses many missed or the lack of relevancies to practical disease transmissions are well-known facts. Even if these claims are accurate, the reviewer failed to see that in the previous version of the manuscript and asked the author to include these in the revision to clearly state the relevancies of their research findings to the infectious disease epidemics. The reviewer wishes the peer review processes can be kept professional instead of personal debates.
Author Response
Thanks for your kindly suggestions. Our reply to all comments are listed below:
Comment 1:
The reviewer fails to see significant improvements in the current form of the manuscript, other than some narrowly opinionated arguments without the supports of reliable references or evidence. The reviewer does believe his comments may not be 100% accurate or unbiased, but the authors can easily establish his/her arguments supported by existing literature works or published historical data. Starting your responses by referring the reviewer with "In general, we believe that the reviewer may not be familiarized with the mechanisms of the transmission of disease during close contact,...," may not be an intelligent move to get positive reviews on your manuscript. The reviewer happens to have more than 20 years of research and practical experience in state and local government agencies. Attack on the reviewer's credentials may not be your best approach to respond to the reviewer's comments.
Response1: We are so sorry if any previous response made the reviewer uncomfortable. We have no any intention for verbal aggression to the reviewer. The reviewer said the study is no obvious relationship with infectious disease transmission last time. We do not agree with this point, because body and head movement/motion, relative position, and face orientation of two speakers, and posture play critical roles in infectious disease transmission. For example, head/body movement, interpersonal distance, and posture determine the exposure of the expired jet of the infected and droplets deposition on surfaces (Qian and Li, 2010; Tang et al., 2011; Richmond-Bryant, 2009). We agree what the reviewer said last time “However, the reviewer is not clear how the results of the study are relevant to the infectious disease transmissions despite the general knowledge on close contacts are related to the disease transmissions.”. In order to make the relationship between our study and infectious disease transmission more clear, we have added some descriptions into the Introduction. Now, we use a paragraph to show the importance of the study to infectious disease transmission. The paragraph is listed below:
“Infection risk via close contact is influenced by interpersonal distance, respiratory activities, and movement of body parts. Interpersonal distance directly affects the risk of virus exposure due to inhalation and deposition, the so-called short-range airborne and large droplet routes, respectively [10]. A threshold distance of close contact less than 1.5 m to 2 m is generally accepted as risky [11-13]. Human respiratory activities such as breathing, talking, and coughing can generate droplets of different numbers and sizes [14-18]. Infectious pathogens are shed and exhaled by the infected during these respiratory activities, and transported by the exhaled air streams, while inhalation of fine droplets and exposure to large droplets are also affected by the inspiratory air streams and body/head/arm movement [19]. Relative face orientation (e.g. face-to-face, face-to-side) and posture are important factors in determining the cross-infection, especially over short distance (Ai et al., 2018). Exposure of face-to-back close contact is much smaller than it of face-to-face pattern (Pantelic et al., 2015; Olmedo et al., 2012). Posture also important in droplet deposition, for example, droplet deposited on trousers out of the thigh of a sitting person should be more than it of a standing person.
Ai, Z.T.; Melikov, A.K. Airborne spread of expiratory droplet nuclei between the occupants of indoor environments: a review. Indoor Air 2018, 28, 500-524.
Olmedo, I.; Nielsen, P.V.; Ruiz de Adana, M.; Jensen, R.L.; Grzelecki, P. Distribution of exhaled contaminants and personal exposure in a room using three different air distribution strategies. Indoor Air 2012, 22, 64-76.
Pantelic, J.; Tham, K.W.; Licina, D. Effectiveness of a personalized ventilation system in reducing personal exposure against directly released simulated cough droplets. Indoor Air 2015, 25, 683-693.
Qian, H.; Li, Y. Removal of exhaled particles by ventilation and deposition in a multibed airborne infection isolation room. Indoor Air 2010, 20, 284-297.
Richmond-Bryant, J. Transport of exhaled particulate matter in airborne infection isolation rooms. Build. Environ. 2009, 44, 44-55.
Tang, J.W.; Noakes, C.J.; Nielsen, P.V.; Eames, I.; Nicolle, A.; Li, Y.; et al. Observing and quantifying airflows in the infection control of aerosol-and airborne-transmitted diseases: an overview of approaches. J. Hosp. Infect. 2011, 77, 213-222.
Comment 2: In general, the reviewer's comments are aimed to have the authors making connections, in the manuscript, between the research results to relevant disease transmissions. The reviewer failed to see enough statements or explanations on these aspects, and respectfully disagree with some of the relevance the previous manuscript claimed. The reviewer was expecting more professional arguments or proof of relevancy with support from the published works of literature and data in this revision, instead of groundless claims.
Response2: As we mentioned above, we have added a paragraph into Introduction and a paragraph into Discussion to show the importance of the study to infectious disease transmission. However, our study is a data-supported research and our main objective is to support data on human behaviors during close contact. Relative position (e.g. interpersonal distance, relative face orientation), head and body motion, and posture are important factors for infection spread via close contact route. We measured all these factors based on more than 400-h indoor data. The data can be used by other researchers for study on infectious disease transmission. The relevancies of our previous research finding to the infectious disease epidemics may not clear, we have added one paragraph into Discussion part (Please see Response 3).
Comment 3:
The authors' responses many missed or the lack of relevancies to practical disease transmissions are well-known facts. Even if these claims are accurate, the reviewer failed to see that in the previous version of the manuscript and asked the author to include these in the revision to clearly state the relevancies of their research findings to the infectious disease epidemics. The reviewer wishes the peer review processes can be kept professional instead of personal debates.
Response3:
We really have no any intention for verbal aggression to the reviewer. We are apologized again if there was any response that made the reviewer feels uncomfortable. There are two objectives of our study. Firstly, our study established a device to detect the human behaviors including indoor position (interpersonal distance), head and body movement, and posture. Secondly, our study improved the temporal resolution of close contact detection from 20 s to 1 s. A temporal resolution of 20 s is not sufficient, as the median value duration of a close contact is 17 s. Our main objective is to support data on human behaviors during close contact. However, according to your suggestion, we also added a paragraph to Discussion to show the relevancies of our research findings to the infectious disease transmission.
The added part in Discussion is listed below:
We collected and analyzed three types of data during close contact: indoor position, head and body motion/movement, and posture. These three factors are important to infectious disease transmission. Indoor position can help calculating the interpersonal distance of people during close contact. The infection risk decreases sharply with the increase of the interpersonal distance (Villafruela et al., 2016; Liu et al., 2017; Olmedo et al., 2013; Ai and Melikov, 2018). Posture, and head and body movement influence the body plume during close contact. For example, short-range exposure can be affected strongly by body plumes (Murakami et al., 2000). Frequent movement of the head and body during conversation can change not only the orientations of the exhaled/inhaled airflows, but also the patterns of body convective flows and the thermal plume. The exhaled airflows of two people also interact with and affect each other (Liu et al., 2017). Various gestures involving small movements of the hands, palms, legs, eyebrows and other small-scale facial features may not significantly affect the body plume or exhaled flows (Pease 1988; Rim and Novoselac, 2009). Posture also important in droplet deposition. For example, more droplets may deposit on thighs of a sitting person if he/she talks with an infected. People have high probability to touch their thighs and legs with the frequency of more than 30 times per hour (Zhang et al., 2018). It may lead to a high infection risk because people also have high touch frequency on mucous membranes (Nicas and Best, 2008). Relative face orientation is a critical factor for exposure during close contact, and it could be calculated by body and head motion. Previous studies found that the exposure during face-to-face close contact is the most, followed by face-to-side pattern, while face-to-back pattern had the lowest exposure (Pantelic et al., 2015; Olmedo et al., 2012).
Ai, Z.T.; Melikov, A.K. Airborne spread of expiratory droplet nuclei between the occupants of indoor environments: a review. Indoor Air 2018, 28, 500-524.
Liu, L.; Li, Y; Nielsen, P.V.; Wei, J.; Jensen, R.L. Short‐range airborne transmission of expiratory droplets between two people. Indoor Air 2017, 27, 452-462.
Murakami, S.; Kato, S.; Zeng, J. Combined simulation of airflow, radiation and moisture transport for heat release from a human body. Build. Environ. 2000, 35, 489-500.
Nicas, M.; Best, D. A study quantifying the hand-to-face contact rate and its potential application to predicting respiratory tract infection. J. Occup. Environ. Hyg. 2008, 5, 347-352.
Olmedo, I.; Nielsen, P.V.; Ruiz de Adana, M.; Jensen, R.L.; Grzelecki, P. Distribution of exhaled contaminants and personal exposure in a room using three different air distribution strategies. Indoor Air 2012, 22, 64-76.
Olmedo, I.; Nielsen, P.V.; Ruiz de Adana, M.; Jensen, R.L. The risk of airborne cross‐infection in a room with vertical low‐velocity ventilation. Indoor Air 2013, 23, 62-73.
Pantelic, J.; Tham, K.W.; Licina, D. Effectiveness of a personalized ventilation system in reducing personal exposure against directly released simulated cough droplets. Indoor Air 2015, 25, 683-693.
Pease, A. Body language: how to read others’ thoughts by their gestures. UK: Sheldon Press; 1988.
Rim, D.; Novoselac, A. Transport of particulate and gaseous pollutants in the vicinity of a human body. Build. Environ. 2009, 44, 1840-1849.
Villafruela, J.M.; Olmedo, I.; San José, J.F. Influence of human breathing modes on airborne cross infection risk. Build. Environ. 2016, 106, 340-351.
Zhang, N.; Li, Y.; Huang, H. Surface touch and its network growth in a graduate student office. Indoor Air 2018, 28, 963-972.
